# Policy Aggregation

**Parand A. Alamdari**[*]
University of Toronto & Vector Institute
parand@cs.toronto.edu

**Soroush Ebadian**[*]
University of Toronto
soroush@cs.toronto.edu

**Ariel D. Procaccia**
Harvard University
arielpro@seas.harvard.edu

## Abstract

We consider the challenge of AI value alignment with multiple individuals that have different reward functions and optimal policies in an underlying Markov decision process. We formalize this problem as one of *policy aggregation*, where the goal is to identify a desirable collective policy. We argue that an approach informed by social choice theory is especially suitable. Our key insight is that social choice methods can be reinterpreted by identifying ordinal preferences with volumes of subsets of the *state-action occupancy polytope*. Building on this insight, we demonstrate that a variety of methods — including approval voting, Borda count, the proportional veto core, and quantile fairness — can be practically applied to policy aggregation.

## 1 Introduction

Early discussion of AI value alignment had often focused on learning desirable behavior from an individual teacher, for example, through inverse reinforcement learning [27, 1]. But, in recent years, the conversation has shifted towards aligning AI models with large groups of people or even entire societies. This shift is exemplified at a policy level by OpenAI's "democratic inputs to AI" program [41] and Meta's citizens' assembly on AI governance [8], and at a technical level by the ubiquity of reinforcement learning from human feedback [30] as a method for fine-tuning large language models.

We formalize the challenge of value alignment with multiple individuals as a problem that we view as fundamental — *policy aggregation*. Our starting point is the common assumption that the environment can be represented as a *Markov decision process (MDP)*. While the states, actions and transition functions are shared by all agents, their reward functions — which incorporate values, priorities or subjective beliefs — may be different. In particular, each agent has its own optimal policy in the underlying MDP. Our question is this: *How should we aggregate the individual policies into a desirable collective policy?*

A naïve answer is to define a new reward function that is the sum of the agents' reward functions (for each state-action pair separately) and compute an optimal policy for this aggregate reward function; such a policy would guarantee maximum *utilitarian social welfare*. This approach has a major shortcoming, however, in that it is sensitive to affine transformations of rewards, so, for example, if we doubled one of the reward functions, the aggregate optimal policy may change. This is an issue because each agent's individual optimal policy is invariant to (positive) affine transformations of rewards, so while it is possible to recover a reward function that induces an agent's optimal policy by

---

[*]Equal contribution. Authors are listed alphabetically.

38th Conference on Neural Information Processing Systems (NeurIPS 2024).

observing their actions over time,[2] it is impossible to distinguish between reward functions that are affine transformations of each other. More broadly, economists and moral philosophers have long been skeptical about *interpersonal comparisons of utility* [19] due to the lack of universal scale — an issue that is especially pertinent in our context. Therefore, aggregation methods that are invariant to affine transformations are strongly preferred.

**Our approach.** To develop such aggregation methods, we look to social choice theory, which typically deals with the aggregation of *ordinal* preferences. To take a canonical example, suppose agents report *rankings* over $m$ alternatives. Under the *Borda count* rule, each voter gives $m - k$ points to the alternative they rank in the $k$'th position, and the alternative with most points overall is selected.

The voting approach can be directly applied to our setting. For each agent, it is (in theory) possible to compute the value of every possible (deterministic) policy, and rank them all by value. Then, any standard voting rule, such as Borda count, can be used to aggregate the rankings over policies and single out a desirable policy. The caveat, of course, is that this method is patently impractical, because the number of policies is exponential in the number of states of the MDP.

The main insight underlying our approach is that ordinal preferences over policies have a much more practical volumetric interpretation in the *state-action occupancy polytope* $\mathcal{O}$. Roughly speaking, a point in the state-action occupancy polytope represents a (stochastic) policy through the frequency it is expected to visit different state-action pairs. If a policy is preferred by an agent to a subset of policies $\mathcal{O}'$, its "rank" is the volume of $\mathcal{O}'$ as a fraction of the volume of $\mathcal{O}$. The "score" of a policy under Borda count, for example, can be interpreted as the sum of these "ranks" over all agents.

**Our results.** We investigate two classes of rules from social choice theory, those that guarantee a notion of fairness and voting rules. By mapping ordinal preferences to the state-action occupancy polytope, we adapt the different rules to the policy aggregation problem.

The former class is examined in Section 5. As a warm-up we start from the notion of *proportional veto core*; it follows from recent work by Chaudhury et al. [7] that a volumetric interpretation of this notion is nonempty and can be computed efficiently. We then turn to *quantile fairness*, which was recently introduced by Babichenko et al. [4]; we prove that the volumetric interpretation of this notion yields guarantees that are far better than those known for the original, discrete setting, and we design a computationally efficient algorithm to optimize those guarantees.

The latter class is examined in Section 6; we focus on volumetric interpretations of $\alpha$-*approval* (including the ubiquitous *plurality* rule, which is the special case of $\alpha = 1$) and the aforementioned Borda count. In contrast to the rules studied in Section 5, existence is a nonissue for these voting rules, but computation is a challenge, and indeed we establish several computational hardness results. To overcome this obstacle, we implement voting rules for policy aggregation through mixed integer linear programming, which leads to practical solutions.

Finally, our experiments in Section 7 evaluate the policies returned by different rules based on their fairness; the results identify quantile fairness as especially appealing. The experiments also illustrate the advantage of our approach over rules that optimize measures of social welfare (which are sensitive to affine transformations of the rewards).

## 2 Related Work

Noothigattu et al. [28] consider a setting related to ours, in that different agents have different reward functions and different policies that must be aggregated. However, they assume that the agents' reward functions are noisy perturbations of a ground-truth reward function, and the goal is to learn an optimal policy according to the ground-truth rewards. In social choice terms, our work is akin to the typical setting where subjective preferences must be aggregated, whereas the work of Noothigattu et al. [28] is conceptually similar to the setting where votes are seen as noisy estimates of a ground-truth ranking [39, 9, 6].

Chaudhury et al. [7] study a problem completely different from ours: fairness in federated learning. However, their technical approach served as an inspiration for ours. Specifically, they consider the proportional veto core and transfer it to the federated learning setting using volumetric arguments, by

---

[2]And we assume this is done accurately, in order to focus on the essence of the policy aggregation problem.

considering volumes of subsets in the space of *models*. Their proof that the proportional veto core is nonempty carries over to our setting, as we explain in Section 5.

There is a body of work on multi-objective reinforcement learning (MORL) and planning that uses a scalarization function to reduce the problem to a single-objective one [32, 21]. Other solutions to MORL focus on developing algorithms to identify a set of policies approximating the problem's Pareto frontier [38]. A line of work more closely related to ours focuses on fairness in sequential decision making, often taking the scalarization approach to aggregate agents' preferences by maximizing a (cardinal) social welfare function, which maps the vector of agent utilities to a single value. Ogryczak et al. [29] and Siddique et al. [34] investigate generalized Gini social welfare, and Mandal and Gan [25], Fan et al. [13] and Ju et al. [23] focus on Nash and egalitarian social welfare. Alamdari et al. [3] study this problem in a non-Markovian setting, where fairness depends on the history of actions over time, and introduce concepts to assess different fairness criteria at varying time intervals. A shortcoming of these solutions is that they are not invariant to affine transformations of rewards — a property that is crucial in our setting, as discussed earlier.

Our work is closely related to the pluralistic alignment literature, aiming to develop AI systems that reflect the values and preferences of diverse individuals [35, 10]. Alamdari et al. [2] propose a framework in reinforcement learning in which an agent learns to act in a way that is considerate of the values and perspectives of humans within a particular environment. Concurrent work explores reinforcement learning from human feedback (RLHF) from a social choice perspective, where the reward model is based on pairwise human preferences, often constructed using the Bradley-Terry model [5]. Zhong et al. [42] consider the maximum Nash and egalitarian welfare solutions, and Swamy et al. [36] propose a method based on maximal lotteries due to Fishburn [14].

## 3 Preliminaries

For $t \in \mathbb{N}$, let $[t] = \{1, 2, \ldots, t\}$. For a closed set $S$, let $\Delta(S)$ denote the probability simplex over the set $S$. We denote the dot product of two vectors as $\langle x, y \rangle = \sum_{i=1}^{d} x_i \cdot y_i$ for $x, y \in \mathbb{R}^d$. A *halfspace* in $\mathbb{R}^d$ determined by $w \in \mathbb{R}^d$ and $b \in \mathbb{R}$ is the set of points satisfying $\{x \in \mathbb{R}^d \mid \langle x, w \rangle \leqslant b\}$. A *polytope* $\mathcal{O} \subseteq \mathbb{R}^d$ is the intersection of a finite number of halfspaces, i.e., a convex subset of the $d$-dimensional space $\mathbb{R}^d$ determined by a set of linear constraints $\{x \mid Ax \leqslant b\}$ where $A \in \mathbb{R}^{k \cdot d}$ is a matrix of coefficients of $k$ linear inequalities and $b \in \mathbb{R}^k$.

### 3.1 Multi-Objective Markov Decision Processes

A multi-objective Markov decision process (MOMDP) is a tuple defined as $M = \langle \mathcal{S}, \mathcal{A}, \mathcal{P}, R_1, \ldots, R_n \rangle$ for the average-reward case and $\langle \mathcal{S}, d_{\text{init}}, \mathcal{A}, \mathcal{P}, R_1, \ldots, R_n, \gamma \rangle$ for the discounted-reward case, where $\mathcal{S}$ is a finite set of states, $\mathcal{A}$ is a finite set of actions, and $\mathcal{P} : (\mathcal{S} \times \mathcal{A}) \to \Delta(\mathcal{S})$ is the transition probability distribution. $\mathcal{P}(s_t, a_t, s_{t+1})$ is the probability of transitioning to state $s_{t+1}$ by taking action $a_t$ in $s_t$. For $i \in [n]$, $R_i : \mathcal{S} \times \mathcal{A} \to \mathbb{R}$ is the reward function of the $i$th agent, the initial state is sampled from $d_{\text{init}} \in \Delta(\mathcal{S})$, and $\gamma \in (0, 1]$ is the discount factor.

A *(Markovian) policy* $\pi(a|s)$ is a probability distribution over the actions $a \in \mathcal{A}$ given the state $s \in \mathcal{S}$. A policy is *deterministic* if at each state $s$ one action is selected with probability of $1$, and otherwise it is *stochastic*. The expected *average* return of agent $i$ for a policy $\pi$ and the expected *discounted* return of agent $i$ for a policy $\pi$ are defined over an *infinite time horizon* as

$$J_i^{\text{avg}}(\pi) = \lim_{T \to \infty} \frac{1}{T} \mathbb{E}_{\pi, \mathcal{P}} \left[ \sum_{t=1}^{T} R_i(s_t, a_t) \right], \quad J_i^{\gamma}(\pi) = (1-\gamma) \mathbb{E}_{\pi, \mathcal{P}} \left[ \sum_{t=1}^{\infty} \gamma^t R_i(s_t, a_t)|_{s_1 \sim d_{\text{init}}} \right]$$

where the expectation is over the state-action pairs at time $t$ based on both the policy $\pi$ and the transition function $\mathcal{P}$.

**Definition 1** (state-action occupancy measure). *Let $\mathcal{P}_\pi^t$ be the probability measure over states at time $t$ under policy $\pi$. The state-action occupancy measure for state $s$ and action $a$ is defined as*

$$d_\pi^{\text{avg}}(s, a) = \lim_{T \to \infty} \frac{1}{T} \mathbb{E} \left[ \sum_{t=1}^{T} \mathcal{P}_\pi^t(s) \pi(a|s) \right], \quad d_\pi^{\gamma}(s, a) = (1 - \gamma) \mathbb{E} \left[ \sum_{t=1}^{\infty} \gamma^t \mathcal{P}_\pi^t(s) \pi(a|s) \right].$$

For both the average and discounted cases, we can rewrite the expected return as the dot product of the state-action occupancy measures and rewards, that is, $J_i(\pi) = \sum_{(s,a)} d_\pi(s, a) \cdot R_i(s, a) =$

$\langle d_\pi, R_i \rangle$. In addition, the policy can be calculated given the occupancy measure as $\pi(a|s) = d_\pi(s, a)/(\sum_a d_\pi(s, a))$ if $\sum_a d_\pi(s, a) > 0$, and $\pi(a|s) = 1/|\mathcal{A}|$ otherwise.

**Definition 2** (state-action occupancy polytope [31, 40]). *For a MOMDP $M$ in the average-reward case, the space of valid state-action occupancies is the polytope*

$$\mathcal{O}^{\mathrm{avg}} = \left\{ d_\pi^{\mathrm{avg}} \mid d_\pi^{\mathrm{avg}} \geqslant 0, \sum_{s,a} d_\pi^{\mathrm{avg}}(s, a) = 1, \sum_a d_\pi^{\mathrm{avg}}(s, a) = \sum_{s', a'} \mathcal{P}(s', a', s) d_\pi^{\mathrm{avg}}(s', a'), \forall s \in \mathcal{S} \right\}.$$

We similarly define this polytope for the discounted-reward case in Appendix A.

A *mechanism* receives a MOMDP and aggregates the agents' preferences into a policy. An economical efficiency axiom in the social choice literature is that of Pareto optimality.

**Definition 3** (Pareto optimality). *For a MOMDP $M$ and $\epsilon \geqslant 0$, a policy $\pi$ is $\epsilon$-Pareto optimal if there does not exist another policy $\pi'$ such that $J_i(\pi') \geqslant J_i(\pi) + \epsilon$ for all $i \in N$, with strict inequality for at least one agent. For $\epsilon = 0$, we simply call such policies Pareto optimal.*

We call a mechanism Pareto optimal if it always returns a Pareto optimal policy. In special cases where all agents unanimously agree on an optimal policy, Pareto optimality implies that the mechanism will return one such policy. We discuss Pareto optimal implementations of all mechanisms in this work.

### 3.2 Voting and Social Choice Functions

In the classical social choice setting, we have a set of $n$ agents and a set $C$ of $m$ alternatives. The preferences of voter $i \in [n]$ is represented as a *strict ordering* over the alternatives $\sigma_i : [m] \rightarrow C$ equal to $\sigma_i(1) \succ_i \sigma_i(2) \succ_i \ldots \succ_i \sigma_i(m)$, where $c_1 \succ_i c_2$ denotes agent $i$ prefers $c_1$ over $c_2$ for $c_1, c_2 \in C$. A (possibly randomized) voting rule aggregates agents' preferences and returns an alternative or a distribution over the alternatives as the collective decision.

**Positional Scoring Rules.** A positional scoring rule with scoring vector $\vec{s} = (s_1, \ldots, s_m)$ such that $s_1 \geqslant s_2 \geqslant \ldots \geqslant s_m$ works as follows. Each agent gives a score of $s_1$ to their top choice, a score of $s_2$ to their second choice, and so on. The votes are tallied and an alternative with the maximum total score is selected. A few of the well-known positional scoring rules are: Plurality: $(1, 0, \ldots, 0)$, Borda: $(m - 1, m - 2, \ldots, 1, 0)$, $k$-approval: $(1, \ldots, 1, 0, \ldots, 0)$ with $k$ ones.

## 4 Occupancy Polytope as the Space of Alternatives

In a MOMDP $M$, each agent $i$ incorporates their values and preferences into their respective reward function $R_i$. Agent $i$ prefers $\pi$ over $\pi'$ if and only if $\pi$ achieves higher expected return, $J_i(\pi) > J_i(\pi')$, and is indifferent between two policies $\pi$ and $\pi'$ if and only if $J_i(\pi) = J_i(\pi')$. As discussed before, given a state-action occupancy measure $d_\pi$ in the state-action occupancy polytope $\mathcal{O}$, we can recover the corresponding policy $\pi$. Therefore, we can interpret $\mathcal{O}$ as the domain of all possible alternatives over which the $n$ agents have heterogeneous *weak* preferences (with ties). Agent $i$ prefers $d_\pi$ to $d_{\pi'}$ in $\mathcal{O}$ if and only if they prefer $\pi$ to $\pi'$. We study the policy aggregation problem through this lens; specifically, we design or adapt voting mechanisms where the (continuous) space of alternatives is $\mathcal{O}$ and agents have weak preferences over them determined by their reward functions $R_i$.

**Affine transformation and reward normalization.** A particular benefit of this interpretation, as mentioned before, is that all positive affine transformations of the reward functions, i.e., $aR_i + b$ for all $a \in \mathbb{R}_{\geqslant 0}$ and $b \in \mathbb{R}$, yield the same weak ordering over the polytope. Hence, we can assume without loss of generality that $J_i(\pi) \in [0, 1]$. Further, we can ignore agents that are indifferent between all policies, i.e., $\min_\pi J_i(\pi) = \max_\pi J_i(\pi)$, and normalize reward functions $R_i \leftarrow \frac{R_i - \min_\pi J_i(\pi)}{\max_\pi J_i(\pi) - \min_\pi J_i(\pi)}$ such that $\min_\pi J_i(\pi) = 0$ and $\max_\pi J_i(\pi) = 1$. The relative ordering of the policies does not change since for all points $d_\pi \in \mathcal{O}$ we have $\sum_{s,a} d_\pi(s, a) = 1$.

**Volumetric definitions.** A major difference between voting over a continuous space of alternatives and the classical voting setting is that the domain of alternatives is infinite and not all voting mechanisms can be directly applied to the policy aggregation problem. In particular, various voting rules require reasoning about the rank of an alternative or the size of some subset of alternatives. For instance, the Borda count of an alternative $c$ over a finite set of alternatives is defined as the number

(or fraction) of candidates ranked below $c$. In the continuous setting, for almost all of the mechanisms that we discuss later, we use the *measure* or *volume* of a subset of alternatives to refer to their size. For a measurable subset $\mathcal{O}' \subseteq \mathcal{O}$, let $\mathrm{vol}(\mathcal{O}')$ denote its measure. The ratio $\mathrm{vol}(\mathcal{O}')/\mathrm{vol}(\mathcal{O})$ is the fraction of alternatives that lie in $\mathcal{O}'$. A probabilistic interpretation is that for a uniform distribution over the polytope $\mathcal{O}$, $\mathrm{vol}(\mathcal{O}')/\mathrm{vol}(\mathcal{O})$ denotes the probability that a policy uniformly sampled from $\mathcal{O}$ lies in $\mathcal{O}'$. We can also define the *expected return distribution* of an agent over $\mathcal{O}$ as a random variable that maps a policy to its expected return, i.e., one that maps $d_\pi \in \mathcal{O}$ to $\langle d_\pi, R_i \rangle$. The pdf and cdf of this r.v. is defined below.

**Definition 4** (expected return distribution). *For a MOMDP $M$ and $v \in \mathbb{R}$, the* expected return distribution *of agent $i \in [n]$ is defined as*

$$f_i(v) = \frac{1}{\mathrm{vol}(\mathcal{O})} \int_{x \in \mathcal{O}} \delta(v - \langle x, R_i \rangle) \, dx, \quad F_i(v) = \int_{x=-\infty}^{v} f_i(x) \, dx = \frac{\mathrm{vol}(\mathcal{O} \cap \{\langle x, R_i \rangle \leqslant v\})}{\mathrm{vol}(\mathcal{O})},$$

*where $f_i$ and $F_i$ are the pdf and cdf of the expected return distribution and $\delta(\cdot)$ is the Dirac delta function indicating $v = \langle x, R_i \rangle$.*

A useful observation about $f_i$, the pdf, is that it is unimodal, i.e., increasing up to its mode $\mathrm{mode}(f_i) \in \arg\max_{v \in \mathbb{R}} f_i(v)$ and decreasing afterwards, which follows from the Brunn–Minkowski inequality [17]. Since $f_i$ (pdf) is the derivative of $F_i$ (cdf), the unimodality of $f_i$ implies that $F_i$ is a convex function in $(-\infty, \mathrm{mode}(f_i)]$ and concave in $[\mathrm{mode}(f_i), \infty)$.

In our algorithms, we use a subroutine that measures the volume of a polytope, which we denote by $\mathrm{vol\text{-}comp}(\{Ax \leqslant b\})$. Dyer et al. [12] designed a fully polynomial time randomized approximation scheme (FPRAS) for computing the volume of polytopes. We report the running time of algorithms in terms of the number of calls to this oracle.

# 5 Fairness in Policy Aggregation

In this section, we utilize the volumetric interpretation of the state-action occupancy polytope to extend fairness notions from social choice to policy aggregation, and we develop algorithms to compute stochastic policies provably satisfying these notions.

## 5.1 Proportional Veto Core

The proportional veto core was first proposed by Moulin [26] in the classical social choice setting with a finite set of alternatives where agents have full (strict) rankings over the alternatives. For simplicity, suppose the number of alternatives $m$ is a multiple of $n$. The idea of the proportional veto core is that $x\%$ of the agents should be able to veto $x\%$ of the alternatives. More precisely, for an alternative $c$ to be in the proportional veto core, there should not exist a coalition $S$ that can "block" $c$ using their proportional veto power of $|S|/n$. $S$ blocks $c$ if they can unanimously suggest $m(1 - |S|/n)$ candidates that they prefer to $c$. For instance, if $c$ is in the proportional veto core, it cannot be the case that a coalition of $60\%$ of the agents unanimously prefer $40\%$ of the alternatives to $c$.

Chaudhury et al. [7] extended this notion to a continuous domain of alternatives in the federated learning setting. We show that such an extension also applies to policy aggregation.

**Definition 5** (proportional veto core). *Let $\epsilon \in (0, 1/n)$. For a coalition of agents $S \subseteq [n]$, let $\mathrm{veto}(S) = |S|/n$ be their veto power. A point $d_\pi \in \mathcal{O}$ is* blocked *by a coalition $S$ if there exists a subset $\mathcal{O}' \subseteq \mathcal{O}$ of measure $\mathrm{vol}(\mathcal{O}')/\mathrm{vol}(\mathcal{O}) \geqslant 1 - \mathrm{veto}(S) + \epsilon$ such that all agents in $S$ prefer all points in $\mathcal{O}'$ to $d_\pi$, i.e., $d_{\pi'} \succ_i d_\pi$ for all $d_{\pi'} \in \mathcal{O}'$ and $i \in S$. A point $d_\pi$ is in the $\epsilon$-proportional veto core if it is not blocked by any coalition.*

A candidate in the proportional veto core satisfies desirable properties that are extensively discussed in prior work [26, 7, 24, 22]. It is worth mentioning that any candidate in the $\epsilon$-proportional veto core, besides the fairness aspect, is also economically efficient as it satisfies $\epsilon$-*Pareto optimality*. This holds since the grand coalition $S = [n]$ can veto any $\epsilon$-Pareto dominated alternative.

Moulin [26] proved that the proportional veto core is nonempty in the discrete setting and Chaudhury et al. [7] proved it for the continuous setting. The following result is a corollary of Theorem 1 of Chaudhury et al. [7].

| **ALGORITHM 1:** Seq. $\epsilon$-Prop. Veto Core [7] | **ALGORITHM 2:** $\epsilon$-Max Quantile Fairness |
|---|---|
| $\mathcal{O}_0 \leftarrow \mathcal{O}, \quad \delta \leftarrow \frac{1}{n} - \frac{\epsilon}{n+1}$ 
 **for** $i = 1$ *to* $n$ **do** 
 $\quad$ Using binary search find $v_i^* \in [0,1]$ s.t. 
 $\quad \quad \text{vol}(\mathcal{O}_{i-1} \cap \{\langle x, R_i \rangle \leqslant v_i^*\}) \approx \delta \cdot \text{vol}(\mathcal{O})$ 
 $\quad \mathcal{O}_i \leftarrow \mathcal{O}_{i-1} \cap \{\langle x, R_i \rangle \geqslant v_i^*\}$ 
 **return** *a welfare maximizing point* $d_\pi \in \mathcal{O}_n$ | **Procedure** $q$-quantile-feasible($q$): 
 $\quad \mathcal{O}_q \leftarrow \{x \in \mathcal{O} \mid \langle x, R_i \rangle \geqslant F_i^{-1}(q), i \in [n]\}$ 
 $\quad$ **if** $\mathcal{O}_q$ *is a feasible linear program* **then return** 
 $\quad \quad$ *True* **else return** *False* 
 Using binary search find maximum $q \in [0,1]$ s.t. 
 $\quad q$-quantile-feasible() is feasible 
 **return** *a welfare maximizing point* $d_\pi \in \mathcal{O}_q$ |

**Theorem 1.** *Let $\epsilon \in (0, 1/n)$. For a policy aggregation problem, the $\epsilon$-proportional veto core is nonempty. Furthermore, such policies can be found in polynomial time using $O(\log(1/\epsilon))$ many calls per agent to* vol-comp.

We refer the reader to the paper of Chaudhury et al. [7] for the complete proof, and provide a high-level description of Algorithm 1, which finds a point in the proportional veto core. Algorithm 1 iterates over the agents and lets the $i$th agent "eliminate" roughly $1/n \cdot \text{vol}(\mathcal{O})$ of the remaining space of alternatives. That is, agent $i$ finds the hyperplane $H_i = \{\langle x, R_i \rangle \leqslant v_i^*\}$ such that its intersection with $O_{i-1}$ (the remaining part of $\mathcal{O}$ at the $i$th iteration) has a volume of approximately $\text{vol}(\mathcal{O})/n$. This value, for each agent, can be found by doing a binary search over $v_i^*$ to a precision of $[v_i^* - \epsilon, v_i^*]$ by $O(\log(1/\epsilon))$ calls to the volume estimating subroutine vol-comp.[3]

**Pareto optimality.** We briefly discuss why Algorithm 1 is Pareto optimal. During the $i$th iteration, the space of policies is contracted by adding a linear constraint of the form $J_i(\pi) \geqslant v_i^*$. If the returned welfare-maximizing policy $\pi$ (derived from $d_\pi$) is Pareto dominated by another policy $\pi'$, then $\pi'$ would satisfy all these linear constraints as $J_i(\pi') \geqslant J_i(\pi) \geqslant v_i^*$ with the earlier inequality being strict for at least one agent. Therefore, $d_{\pi'} \in \mathcal{O}_n$ and $\pi'$ achieves a higher social welfare, which is a contradiction. The same argument can be used to establish the Pareto optimality of other mechanisms discussed later; each of these mechanisms searches for a policy that satisfies certain lower bounds on agents' utilities, from which a welfare-maximizing, Pareto optimal policy can be selected.

## 5.2 Quantile Fairness

Next, we consider an egalitarian type of fairness based on the occupancy polytope, building on very recent work by Babichenko et al. [4] in the discrete setting; surprisingly, we show that it is possible to obtain stronger guarantees in the continuous setting.

Babichenko et al. [4] focus on the fair allocation of a set of $m$ indivisible items among $n$ agents, where each item must be fully allocated to a single agent. They quantify the extent an allocation $A$ is *fair* to an agent $i$ by the fraction of allocations over which $i$ prefers $A$ (note that the number of all discrete allocations is $n^m$). In other words, if one randomly samples an allocation, the fairness is measured by the probability that $A$ is preferred to the random allocation. An allocations is $q$-*quantile fair* for $q \in [0,1]$ if *all* agents consider this allocation among their top $q$-quantile allocations. Babichenko et al. [4] aim to find a universal value of $q$ such that for any fair division instance, a $q$-quantile fair allocation exists. They make an interesting connection between $q$-quantile fair allocations and the Erdős Matching Conjecture, and show under the assumption that the conjecture holds, $(1/2e)$-quantile fair allocations exist for all instances.[4]

We ask the same question for policy aggregation, and again, a key difference is that our domain of alternatives is continuous. The notion of $q$-quantile fairness extends well to our setting. Agents assess the fairness of a policy $\pi$ based on the fraction of the occupancy polytope (i.e., the set of all policies) to which they prefer $\pi$, or equivalently, the probability that they prefer the chosen policy to a randomly sampled policy.

**Definition 6** ($q$-quantile fairness). *For a MOMDP $M$ and $q \in [0,1]$, a policy $\pi$ is $q$-quantile fair if for every agent $i \in [n]$, $\pi$ is among $i$'s top $q$-fraction of policies,* $\text{vol}(\mathcal{O} \cap \{\langle x, R_i \rangle \leqslant J_i(\pi)\}) \geqslant q \cdot \text{vol}(\mathcal{O})$.

---

[3]It is important to check the non-emptyness of the search space $\mathcal{O}_i$ as an over-estimation of $v_i^*$ could lead to an empty feasible region.

[4]This result is for arbitrary monotone valuations. For additive valuations, they show $\frac{0.14}{e}$-quantile fair allocations always exist without any assumptions.

We show that a $(1/e)$-quantile fair policy always exist and that this ratio is tight; note that this bound is twice as good as that of Babichenko et al. [4], and it is unconditional. We prove this by making a connection to an inequality due to Grünbaum [17]. The *centroid* of a polytope $P$ is defined as $\text{centroid}(P) = \frac{\int_{x \in P} x \, dx}{\int_{x \in P} 1 \, dx}$.

**Lemma 1** (Grunbaum's Inequality). *Let $P$ be a polytope and $w$ a direction in $\mathbb{R}^d$. For the halfspace $H = \{x \mid \langle w, x - \text{centroid}(P) \rangle \geqslant 0\}$, it holds that*

$$\frac{1}{e} \leqslant \left(\frac{d}{d+1}\right)^d \leqslant \frac{\text{vol}(P \cap H)}{\text{vol}(P)} \leqslant 1 - \left(\frac{d}{d+1}\right)^d \leqslant 1 - \frac{1}{e},$$

*Furthermore, this is tight for the $n$-dimensional simplex.*

**Theorem 2.** *For every MOMDP $M$, there always exist $q$-quantile fair policy where $q = (\frac{\ell-1}{\ell})^{\ell-1}$ and $\ell = |\mathcal{S}| \cdot |\mathcal{A}|$. Note that $q \geqslant \frac{1}{e} \approx 36.7\%$. Furthermore, this bound is tight: For any $\ell$, there is an instance with a single state and $\ell$ actions where no $q$-quantile fair policy exists for any $q > (\frac{\ell-1}{\ell})^{\ell-1}$.*

*Proof.* First, we show that the centroid of the occupancy polytope $c = \text{centroid}(\mathcal{O})$ is $q$-quantile fair policy for the aforementioned $q$. Since $\mathcal{O}$ is a subset of the $(n-1)$-simplex (see Definition 2), $\mathcal{O}$ has a nonzero volume in some lower dimensional space $\ell' \leqslant |\mathcal{S}| \cdot |\mathcal{A}| - 1$. By invoking Grunbaum's inequality (Lemma 1) with $w_i$ being equal to $R_i$ projected to the $\ell'$-dimensional subspace for all agents $i \in [n]$, we have that $\text{vol}(\mathcal{O} \cap H_i) \geqslant (\frac{\ell'}{\ell'+1})^{\ell'} \cdot \text{vol}(\mathcal{O})$ where $H_i = \{\langle x - c, w_i \rangle \geqslant 0\} = \{\langle x, R_i \rangle \geqslant J_i(c)\}$. Since $\ell' \leqslant \ell - 1$, we have $(\frac{\ell'}{\ell'+1})^{\ell'} \geqslant (\frac{\ell-1}{\ell})^{\ell-1}$, which completes the proof.

To show tightness, take a MOMDP with a single state $\mathcal{S} = \{s\}$ — hence a constant transition function $\mathcal{P}(\cdot) = s$ — and $\ell$ actions $\mathcal{A} = \{a_1, \dots, a_\ell\}$ and $\ell$ agents. The reward function of agent $i$ is $R_i(s, a_i) = 1$ and $0$ otherwise. The state-action occupancy polytope of this MOMDP is the $(\ell-1)$-dimensional simplex $\mathcal{O} = \{\sum_a d_\pi(s, a) = 1, d_\pi \in \mathbb{R}_{\geqslant 0}^{1 \times \ell}\}$. Take any point in $d_\pi \in \mathcal{O}$. There exists at least one agent $i$ that has $J_i(d_\pi) = d_\pi(s, a_i) \leqslant \frac{1}{\ell}$. Take the halfspace $H_i = \{\langle x, R_i \rangle \geqslant \frac{1}{\ell}\}$. Observe that $H_i \cap \mathcal{O}$ is equivalent to a smaller $(\ell-1)$-dimensional simplex $\{\sum_a d_\pi(s, a) = 1 - \frac{1}{\ell}\}$ which has volume of $\text{vol}\left((1 - \frac{1}{\ell})\mathcal{O}\right) = \left(\frac{\ell-1}{\ell}\right)^{\ell-1} \text{vol}(\mathcal{O})$. Therefore, $\frac{\text{vol}(\mathcal{O} \cap H_i)}{\text{vol}(\mathcal{O})} = \left(\frac{\ell-1}{\ell}\right)^{\ell-1}$. □

The centroid of the occupancy polytope, as per Theorem 2, attains a worst-case measure of quantile-fairness. However, the centroid policy can be highly suboptimal as it disregards the preferences of the agents involved. For instance, there could exist a $99\%$-quantile fair policy. To this end, we take an egalitarian approach and aim to find a $q$-quantile fair policy with the maximum $q$.

**Max quantile fair algorithm.** Algorithm 2 searches for the optimal value $q^*$, for which a $q^*$-quantile fair policy exists, and gets close to $q^*$ by a binary search. To perform the search, we need a subroutine that checks, for a given value of $q$, if a $q$-quantile fair policy exists. Suppose we have the $q$-quantile expected return $F_i^{-1}(q)$ for all $i$, that is, the expected return amount $v_i$ such that $F_i(v_i) = q$. Then, the problem of existence of a $q$-quantile fair policy is equivalent to the feasibility of the linear program $\{x \in \mathcal{O} \mid \langle x, R_i \rangle \geqslant F_i^{-1}(q), i \in [n]\}$, which can be solved in polynomial time. Importantly, after finding a good approximation of $q$, there can be infinitely many policies that are $q$-quantile fair and there are various ways to select the final policy after finding the value $q$. As mentioned earlier, a desirable efficiency property is Pareto optimality; to satisfy it, one can return the policy that maximizes the sum of agents' expected returns among the ones that are $q$-quantile fair. Finally, to calculate $F_i^{-1}(q)$, we can again use binary search to get $\delta$ close to the value $v_i$ for which $F_i(v_i) = q$ using $O(\log 1/\delta)$ calls to vol-comp. The discussion above is summarized below.

**Proposition 3.** *Assuming an optimal oracle for $F_i^{-1}$, Algorithm 2 finds a $q$-quantile fair policy that is $\epsilon$ close to the optimal value in polynomial time with $O(\log(1/\epsilon))$ per agent calls to the oracle. A $\delta$-approximation to $F_i^{-1}(q)$ can be computed using $O(\log(1/\delta))$ calls to vol-comp.*

## 6   Policy Aggregation with Voting Rules

In this section, we adapt existing voting rules from the discrete setting to policy aggregation and discuss their computational complexity.

| **ALGORITHM 3:** $\alpha$-Approvals MILP | **ALGORITHM 4:** $\epsilon$-Borda count MILP |
|---|---|
| compute $F_i^{-1}(\alpha)$ for all $i \in [n]$
solve the mixed integer linear program

$$\max \sum_{i \in [n]} a_i$$

s.t. $\quad a_i \cdot F_i^{-1}(\alpha) \leqslant \langle d_\pi, R_i \rangle \quad \forall i \in [n]$
$\quad\quad d_\pi \in \mathcal{O}, \quad a_i \in \{0,1\} \quad\quad \forall i \in [n]$

**return** a Pareto optimal policy subject
to max $\alpha$-approval $\{i \mid a_i = 1\}$ | compute $F_i^{-1}(k\epsilon)$ for all $i \in [n]$ and $k \in \left[\frac{1}{\epsilon}\right]$
solve the mixed integer linear program

$$\max \sum_{i \in [n]} \sum_{k \in [1/\epsilon]} a_{i,k} \cdot (F_i(k\epsilon) - F_i((k-1)\epsilon))$$

s.t. $\quad a_{i,k} \cdot k\epsilon \leqslant \langle d_\pi, R_i \rangle \quad \forall i \in [n]$
$\quad\quad d_\pi \in \mathcal{O}, \quad a_{i,k} \in \{0,1\}, \quad\quad \forall i \in [n], k \in [1/\epsilon]$

**return** a Pareto optimal policy subject to max Borda score |

**Plurality.** The plurality winner is the policy that achieves the maximum number of plurality votes or "approvals," where agent $i$ approves a policy $\pi$ if it achieves their maximum expected return $J_i(\pi) = \max_{\pi'} J_i(\pi') = 1$. Hence the plurality winner is a policy in $\arg\max_\pi \sum_{i \in [n]} \mathbb{I}[J_i(\pi) = 1]$. This formulation does not require the volumetric interpretation. However, in contrast to the discrete setting where one can easily count the approvals for all candidates, we show that solving this problem in the context of policy aggregation is not only **NP**-hard, but hard to approximate up to factor of a $1/n^{1/2-\epsilon}$. We establish the hardness of approximation by a reduction from the *maximum independent set* problem [20]; we defer the proof to Appendix B.

**Theorem 4.** *For any fixed $\epsilon \in (0,1)$, there is no polynomial-time $\frac{1}{n^{1/2-\epsilon}}$-approximation algorithm for the maximum plurality score unless $\mathbf{P} = \mathbf{NP}$.*

Nevertheless, we can compute plurality in practice, as we discuss below.

$\alpha$**-approval.** We extend the $k$-approval rule using the volumetric interpretation of the occupancy polytope, similarly to the $q$-quantile fairness definition. For some $\alpha \in [0,1]$, agents approve a policy $\pi$ if its return is among their top $\alpha$ fraction of $\mathcal{O}$, i.e., $F_i(J_i(\pi)) \geqslant \alpha$. The $\alpha$-approval winner is a policy that has the highest number of $\alpha$-approvals, so it is in $\arg\max_\pi \sum_{i \in [n]} \mathbb{I}[F_i(J_i(\pi)) \geqslant \alpha]$. Note that plurality is equivalent to 1-approval. It is worth mentioning that there can be infinitely many policies that have the maximum approval score and, to avoid a suboptimal decision, one can return a Pareto optimal solution among the set of $\alpha$-approval winner policies.

Theorem 2 shows that for $\alpha \leqslant 1/e$, there always exists a policy that all agents approve, and by Proposition 3 such policies can be found in polynomial time, assuming access to an oracle for volumetric computations. Therefore, the problem of finding an $\alpha$-approval winner is "easy" for $\alpha \in (0, 1/e)$. In sharp contrast, for $\alpha = 1$ — namely, plurality — Theorem 4 gives a hardness of approximation. The next theorem shows the hardness of computing $\alpha$-approval for $\alpha \in (7/8, 1]$ via a reduction from the MAX-2SAT problem. We defer the proof to Appendix B.

**Theorem 5.** *For $\alpha \in (7/8, 1]$, computing a policy with the highest $\alpha$-approval score is $\mathbf{NP}$-hard. This even holds for binary reward vectors and when every $F_i$ has a closed form.*

Given the above hardness result, to compute the $\alpha$-approval rule, we turn to *mixed-integer linear programming (MILP)*. Algorithm 3 simply creates $n$ binary variables for each agent $i$ indicating whether $i$ $\alpha$-approves the policy, i.e., $F_i(J_i(\pi)) \geqslant \alpha$ which is equivalent to $J_i(\pi) \geqslant F_i^{-1}(\alpha)$. To encode the expected return requirement for agent $i$ to approve a policy as a linear constraint, we precompute $F_i^{-1}(\alpha)$. This can be done by a binary search similar to Algorithm 2. Importantly, Algorithm 3 has one binary variable per agent and only $n$ constraints which is key to its practicability.

**Borda count.** The Borda count rule also has a natural definition in the continuous setting. In the discrete setting, the Borda score of agent $i$ for alternative $c$ is the number of alternatives $c'$ such that $c \succ_i c'$. In the continuous setting, $F_i(J_i(\pi))$ indicates the volume of the occupancy polytope to which agent $i$ prefers $\pi$. The Borda count rule then selects a policy among $\arg\max_\pi \sum_{i \in [n]} F_i(J_i(\pi))$.

The computational complexity of the Borda count rule remains an interesting open question, though we make progress on two fronts.[5] First, we identify a sufficient condition under which we can find an approximate max Borda count policy using convex optimization in polynomial time. Second, similar to Algorithm 3, we present a MILP to approximate the Borda count rule in Algorithm 4.

---

[5] We suspect the problem to be $\mathbf{NP}$-hard since the objective resembles a summation of "sigmoidal" functions over a convex domain, which is known to be $\mathbf{NP}$-hard [37].

The first is based on the observation in Section 4 that $F_i$ is concave in range $[\text{mode}(f_i), \infty)$. We assume that the max Borda count policy $\pi$ appears in the concave portion of all agents, i.e., $J_i(\pi) \geqslant \text{mode}(f_i)$ for all $i \in [n]$. Then, the problem becomes a maximization of the concave objective $\max \sum_i F_i(\langle d_\pi, R_i \rangle)$ over the convex domain $\{d_\pi \in \mathcal{O} \mid \langle d_\pi, R_i \rangle \geqslant \text{mode}(f_i), \forall i \in [n]\}$.

Second, Algorithm 4 is a MILP that finds an approximate max Borda count policy. As a pre-processing step, we estimate $F_i$ for each agent $i$ separately. We measure $F_i$ for the fixed expected return values of $\{\epsilon, 2\epsilon, \ldots, 1-\epsilon, 1\}$. This accounts for $1/\epsilon$ oracle calls to vol-comp per agent. Then, for the MILP, we introduce $1/\epsilon$ binary variables for each agent indicating their $\epsilon$-rounded return levels, i.e., $a_{i,k} = 1$ iff $\langle d_\pi, R_i \rangle \geqslant k\epsilon$ for $k \in [1/\epsilon]$. The MILP then searches for an occupancy measure $d_\pi \in \mathcal{O}$ with maximum total Borda score among the $\epsilon$-rounded expected return vectors (see Appendix C for more details).

Finally, we make a novel connection between $q$-quantile fairness and Borda count in Theorem 6. We defer the proof to Appendix B. A corollary of Theorems 2 and 6 is that the policy returned by $\epsilon$-max quantile fair algorithm (Algorithm 2) achieves a $(1/e - \epsilon)$ multiplicative approximation of the maximum Borda score.

**Theorem 6.** *A $q$-quantile fair policy is a $q$-approximation of the maximum Borda score.*

## 7   Experiments

**Environment.** We adapt the dynamic attention allocation environment introduced by D'Amour et al. [11]. We aim to monitor several sites and prevent potential incidents, but limited resources prevent us from monitoring all sites at all times; this is inspired by applications such as food inspection and pest control [6]. There are $m = 5$ warehouses and each can be in 3 different stages: normal (norm), risky (risk) and incident (inc). There are $|\mathcal{S}| = 3^m$ states containing all possible stages of all warehouses. In each step, we can monitor at most one site, so there are $m + 1$ actions, where action $m + 1$ is no operation and action $i \leqslant m$ is monitoring warehouse $i$. There are $n$ agents; each agent $i$ has a list $\ell_i$ of warehouses that they consider valuable and a reward function $R_i$. In each step $t$, $R_i(s_t, a_t) = -\mathbb{I}[a_t \leqslant m] - \sum_{j \in \ell_i} \rho_i w_j \cdot \mathbb{I}[s_{t,j} = \text{inc} \wedge a_t \neq j]$, where $w_j \in \{100, 150, \cdots, 250\}$ denotes the penalty of an incident occurring in warehouse $j$, $\rho_i$ is the scale of penalties for agent $i$ which is sampled from $\{0.25, 0.5, \cdots, n\}$, and $-1$ is the cost of monitoring. In each step, if we monitor warehouse $j$, its stage becomes normal. If not, it changes from norm to risk and from risk to inc with probabilities $p_{j,\text{risk}}$ and $p_{j,\text{inc}}$, and stays the same otherwise. Probabilities are sampled i.i.d. uniformly from $[0.5, 0.8]$. The state transitions $\mathcal{P}$ is the product of the warehouses' stage transitions.

**Rules.** We compare the outcomes of policy aggregation with different rules: *max-quantile, Borda, $\alpha$-approval* ($\alpha = 0.9, 0.8$), *egalitarian* (maximize minimum return) and *utilitarian* (maximize sum of returns). We sample $5 \cdot 10^5$ random policies based on which we fit a generalized logistic function to estimate the cdf of the expected return distribution $F_i$ (Definition 4) for every agent. The policies for $\alpha$-approval voting rules are optimized with respect to maximum utilitarian welfare. The egalitarian rule finds a policy that maximizes the expected return of the worst-off agent, then optimizes for the second worst-off agent, and so on. The implementation details of Borda count are in Appendix D.

**Results.** In Figure 1, we report the normalized expected return of agents as $\frac{J_i(\pi) - \min_\pi J_i(\pi)}{\max_{\pi'} J_i(\pi') - \min_{\pi''} J_i(\pi'')}$ (sorted from lowest to highest) which are averaged over 10 different environment and agents instances. We observe that the utilitarian and egalitarian rules are sensitive to the different agents' reward scales and tend to perform unfairly. The utilitarian rule achieves the highest utilitarian welfare by almost ignoring one agent. The egalitarian rule achieves higher return for the worst-off agents compared to the utilitarian rule, but still yields an inequitable outcome. The max-quantile rule tends to return the fairest outcomes with similar normalized returns for the agents. The Borda rule, while not a fair rule by design, tends to find fair outcomes which are slightly worse than the max-quantile rule. The $\alpha$-approval rule with max utilitarian completion tends to the utilitarian rule as $\alpha \to 0$ and to plurality as $\alpha \to 1$. Importantly, although not shown in the plots, the plurality rule ignores almost all agents and performs optimally for a randomly selected agent.

In addition to the fine-grained utility distributions, in Table 1, we report two aggregate measures based on agents' utilities: (i) the Gini index, a statistical measure of dispersion defined

---

[6]The code for the experiments is available at https://github.com/praal/policy-aggregation.

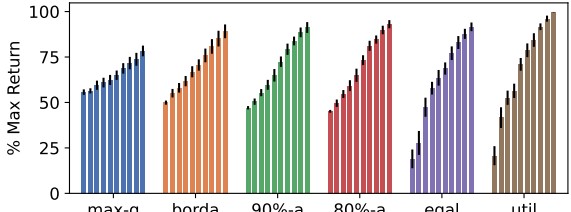
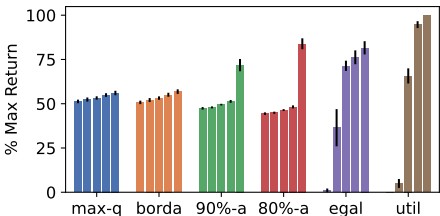

(a) 10 agents consider a random subset of warehouses valuable     (b) 5 symmetric agents, one per warehouse

Figure 1: Comparison of policies optimized by different rules in two different scenarios based on the normalized expected return for agents. The bars, grouped by rule, correspond to agents sorted based on their normalized expected return. The error bars show the standard error of the mean.

| Rules | 5 symmetric agents, one per warehouse | | 10 agents, random subsets of warehouses | |
|---|---|---|---|---|
| | Gini index | Nash welfare | Gini index | Nash welfare |
| egalitarian | $0.2864 \pm 0.0295$ | $0.2208 \pm 0.0717$ | $0.2126 \pm 0.0209$ | $0.4655 \pm 0.0581$ |
| utilitarian | $0.4392 \pm 0.0094$ | $0.0502 \pm 0.0174$ | $0.2020 \pm 0.0182$ | $0.5736 \pm 0.0471$ |
| 80%-approvals | $0.1233 \pm 0.0047$ | $0.5186 \pm 0.0051$ | $0.1352 \pm 0.0037$ | $0.6741 \pm 0.0200$ |
| 90%-approvals | $0.0793 \pm 0.0056$ | $0.5286 \pm 0.0053$ | $0.1257 \pm 0.0034$ | $0.6746 \pm 0.0211$ |
| Borda | $0.0225 \pm 0.0024$ | $0.5356 \pm 0.0062$ | $0.1029 \pm 0.0083$ | $0.6801 \pm 0.0261$ |
| max-quantile | $0.0188 \pm 0.0022$ | $0.5355 \pm 0.0062$ | $0.0625 \pm 0.0067$ | $0.6474 \pm 0.0232$ |

Table 1: Comparison of policies optimized by different rules in two scenarios based on Gini index and Nash welfare based on their normalized expected return averaged. We report the mean and the standard error.

as $\frac{\sum_{i \in N} \sum_{j \in N} |J_i(\pi) - J_j(\pi)|}{2n \sum_{i \in N} J_i(\pi)}$ — where a lower Gini index indicates a more equitable distribution, and (ii) the Nash welfare, defined as the geometric mean of agents' utilities $\left(\prod_{i \in N} J_i(\pi)\right)^{1/n}$ — where a higher Nash welfare is preferable. We observe a similar trend as above, where utilitarian and egalitarian rules perform worse across both metrics. For the other four rules, the Nash welfare scores are comparable in both scenarios, with Borda showing slightly better performance. The Gini index, however, highlights a clearer distinction among the rules, with max-quantile performing better.

## 8   Discussion

We conclude by discussing some of the limitations of our approach. A first potential limitation is computation. When we started our investigation of the policy aggregation problem, we were skeptical that ordinal solutions from social choice could be practically applied. We believe that our results successfully lay this initial concern to rest. However, additional algorithmic advances are needed to scale our approach beyond thousands of agents, states, and actions. Additionally, an interesting future direction is to apply these rules within continuous state or action spaces, as well as in online reinforcement learning setting where the environment remains unknown.

A second limitation is the possibility of strategic behavior. The Gibbard-Satterthwaite Theorem [16, 33] precludes the existence of "reasonable" voting rules that are strategyproof, in the sense that agents cannot gain from misreporting their ordinal preferences; we conjecture that a similar result holds for policy aggregation in our framework. However, if reward functions are obtained through inverse reinforcement learning, successful manipulation would be difficult: an agent would have to act in a way that the learned reward function induces ordinal (volumetric) preferences leading to a higher-return aggregate stochastic policy. This separation between the actions taken by an agent and the preferences they induce would likely alleviate the theoretical susceptibility of our methods to strategic behavior.

## Acknowledgments

We gratefully acknowledge funding from the Natural Sciences and Engineering Research Council of Canada (NSERC) and the Canada CIFAR AI Chairs Program (Vector Institute). This work was also partially supported by the Cooperative AI Foundation; by the National Science Foundation under grants IIS-2147187, IIS-2229881 and CCF-2007080; and by the Office of Naval Research under grants N00014-20-1-2488 and N00014-24-1-2704.

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

# A  State-action Occupancy Polytope for Discounted Reward

**Definition 7** (State-action Occupancy Polytope for discounted-reward [31, 40]). *For a MOMDP $M$ in the discounted-reward case, the space of valid state-action occupancies is the polytope*

$$\mathcal{O}^\gamma = \left\{ d_\pi^\gamma \mid d_\pi^\gamma \geqslant 0, \sum_a d_\pi^\gamma(s, a) = (1 - \gamma)d_{\text{init}}(s) + \gamma \sum_{s', a'} \mathcal{P}(s', a', s)d_\pi^\gamma(s', a'), \forall s \in \mathcal{S} \right\}$$

# B  Proofs of Section 6

We define a *fully connected MOMDP* as a MOMDP $M$ where for every pair of states $s, s' \in \mathcal{S}$ and any action $a \in \mathcal{A}$, $\mathcal{P}(s, a, s') = \frac{1}{|\mathcal{S}|}$. In all the hardness proofs, we create a fully connected MOMDP. For such MOMDPs, it is not difficult to observe that the state-action occupancy polytope, for both the average and discounted reward, is equivalent to $\{d_\pi \mid \sum_a d_\pi(s, a) = \frac{1}{|\mathcal{S}|}, \forall s \in \mathcal{S}\}$ for both the discounted and average reward case. Furthermore,

$$\pi(a|s) = \frac{d_\pi(s, a)}{\sum_{a' \in \mathcal{A}} d_\pi(s, a')} = |\mathcal{S}| \cdot d_\pi(s, a) \tag{1}$$

## B.1  Proof of Theorem 4

*Proof of Theorem 4.* We show hardness of approximation by a approximation-preserving reduction from the *maximum independent set (MIS)* problem.

**Definition 8** (maximum independent set (MIS)). *For a graph $G = (V, E)$ with vertex set $V$ and edge set $E \subseteq 2^{\binom{V}{2}}$, the* maximum independent set (MIS) *of $G$ is the maximum subset of vertices $V' \subseteq V$ such that there are no edges between any pair of vertices $v_1, v_2 \in V'$.*

**Theorem 7** (Håstad [20]). *For any fixed $\epsilon \in (0, 1)$, there is no polynomial-time $\frac{1}{n^{1/2-\epsilon}}$-approximation algorithm for the MIS problem unless $\mathbf{P} = \mathbf{NP}$, and no $\frac{1}{n^{1-\epsilon}}$-approximation algorithm unless $\mathbf{ZPP} = \mathbf{NP}$.*

**Construction of MOMDP.** Let $G = ([n], E)$ be a graph for which we want to find the maximum independent set. Create a fully connected MOMDP $M$ with $|E|$ states $\{s_e\}$, one per edge $e \in E$. There are only two actions $\mathcal{A} = \{a_1, a_2\}$. In state $s_e$ of edge $e = (e_1, e_2)$, performing action $a_1$ and $a_2$ correspond to $e_1$ and $e_2$ respectively. We create $n$ agents where agent $i$ corresponds to vertex $i$. The reward function of agent $i$ for state $s_e$ and action $a_k$ for $k \in \{1, 2\}$ is defined as

$$R_i(s_e, a_k) = \begin{cases} 1, & \text{if } e_k = i, \\ 0, & \text{o.w.} \end{cases}$$

In other words, the reward functions encode the set of edges incident to vertex $i$.

**Correctness of reduction.** A policy $\pi$ is optimal for agent $i$ iff for all the edges incident to $i$ the action corresponding to $i$ is selected with probability 1. If a policy $\pi$ is considered optimal by two agents $i$ and $i'$, then $e = (i, i') \notin E$, since at state $s_e$ either $\pi(a_1|s) = 1$ or $\pi(a_2|s) = 1$. Therefore, the set of agents that consider a policy optimal corresponds to an independent set in $G$. Furthermore, take any independent set $V' \subseteq [n]$. Let $\pi$ be the policy that for each edge $e$ selects the action corresponding to the vertex in $V'$ and, if no such vertices exist, select one arbitrarily. This policy is well defined since $V'$ is an independent set and there are no edges with both vertices from $V'$. Policy $\pi$ is optimal for agents of $V'$ since at each state their favourite action is selected. Thus, we have an equivalence between the maximum independent set of $G$ and the plurality winner policy of $M$. Therefore, the hardness of approximation follows from Theorem 7. □

## B.2  Proof of Theorem 5

*Proof of Theorem 5.* We reduce the MAX-2SAT problem to finding an $\alpha$-approval policy for $\alpha \in (7/8, 1]$.

For two Boolean variables $x_1$ and $x_2$, we denote the disjunction (i.e., logical or) of two variables by $x_1 \vee x_2$ and the negation of $x_1$ by $\neg x_1$.

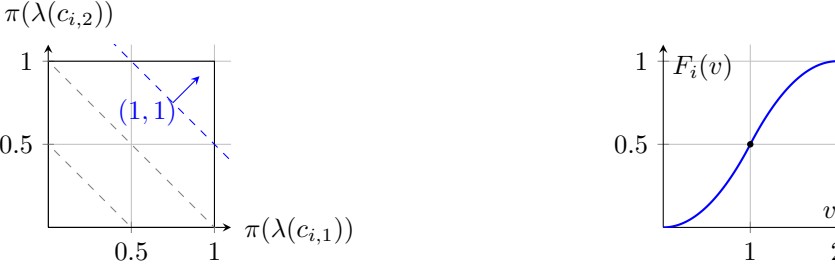

(a) The scaled occupancy polytope        (b) The expected return cdf

Figure 2: The effective state-action occupancy polytope of agents and their expected return distribution.

**Definition 9** (maximum 2-satisfiability (MAX-2SAT))**.** *Given a set of $m$ Boolean variables* $\{x_1, \ldots x_m\}$ *and a set of $n$ 2-literal disjunction clauses* $\{C_1, \ldots, C_n\}$*, the goal of the* maximum 2-satifiability problem (MAX-2SAT) *is to find the maximum number of clauses that can be satisfied together by an assignment* $\phi : \{x_j\}_{j \in [n]} \to \{\text{True}, \text{False}\}$*.*

Garey et al. [15] showed that the MAX-2SAT problem is NP-hard.

**Construction of MOMDP.** For an instance of the MAX-2SAT problem, let $\{C_1, \ldots, C_n\}$ be a set of $m$ 2-literal disjunction clauses over $n$ variables $\{x_1, \ldots x_m\}$. We create a fully connected MOMDP $M$ with $m$ states — state $s_j$ representing variable $x_j$. There are only two actions, $a_{\text{True}}$ and $a_{\text{False}}$ which at state $s_j$ correspond to setting variable $x_j$ to True and False respectively. This way, a policy $\pi(a_{\text{True}}|s_j)$ can be interpreted as the probability of setting the variable $x_j$ to True, subject to $\pi(a_{\text{True}}|s_j) = 1 - \pi(a_{\text{False}}|s_j)$.

**Agents and reward function.** We introduce some notation before introducing the agents and their reward function. Take a clause $C_i = c_{i,1} \vee c_{i,2}$ where $c_{i,k} \in \{x_j, \neg x_j\}_{j \in [m]}$ for $k \in [2]$. By combining the former relation with the mapping of $x_j$ and $\neg x_j$ to state-action pairs $(s_j, a_{\text{True}})$ and $(s_j, a_{\text{False}})$ respectively, we define $\lambda : \{c_{i,1}, c_{i,2}\} \to \mathcal{S} \times \mathcal{A}$. For every $c_{i,k}$, $\lambda(c_{i,k})$ is the state-action pair that evaluates $c_{i,k}$ to True when selected with probability one. Now, we are ready to introduce the agents. For each clause $C_i$, we create 3 agents per clause as follows:

- Agent $\text{ag}_{i,1}$ with a reward function that is 1 for $\lambda(c_{i,1})$ and $\lambda(c_{i,2})$ and zero otherwise. An optimal policy of this agent selects the two state-action pairs with probability one, which can be interpreted as $c_{i,1} \leftarrow \text{True}$ and $c_{i,2} \leftarrow \text{True}$ that implies $C_i = \text{True}$.

- Agent $\text{ag}_{i,2}$ with a reward function that is 1 for $\lambda(c_{i,1})$ and $\lambda(\neg c_{i,2})$ — note the negation. This is another assignment of variables, $c_{i,1} \leftarrow \text{True}$ and $c_{i,2} \leftarrow \text{False}$, that implies $C_i \leftarrow \text{True}$.

- Similarly, the reward function of $\text{ag}_{i,3}$ is 1 for $\lambda(\neg c_{i,1})$ and $\lambda(c_{i,2})$ (which again implies $C_i \leftarrow \text{True}$).

For ease of exposition, and by a slight abuse of notation, in the rest of the proof we use $\pi$ to refer to the occupancy measure. From Equation (1) we have $|\mathcal{S}| \cdot d_\pi(s, a) = \pi(a|s)$ and the $\alpha$-approval rule is invariant to affine transformation. Therefore, we let $J_i(\pi) = \langle \pi, R \rangle$.

**Expected return distribution.** The expected return of agent $\text{ag}_{i,1}$ for a policy $\pi$ is $\pi(\lambda(c_{i,1})) + \pi(\lambda(c_{i,2}))$. For conciseness, let $v_1 = \pi(\lambda(c_{i,1})), v_2 = \pi(\lambda(c_{i,2}))$. Then, the cdf is

$$F_{\text{ag}_{i,1}}(v) = \int_{v_1=0}^{v} \int_{v_2=0}^{v} \mathbb{I}[v_1 + v_2 \leqslant v] \cdot dv_1 \, dv_2 = \begin{cases} \frac{v^2}{2} & \text{for } v \in [0, 1], \\ 1 - \frac{(2-v)^2}{2} & \text{for } v \in [1, 2]. \end{cases}$$

See Figure 2 for a visualization. The cdf of all other agents have the same form. For each of the $3n$ agents above, their maximum expected return is 2.

**Rounding a policy.** Observe that $F_i(3/2) = 7/8$. For agent $\text{ag}_{i,1}$ to $\alpha$-approve a policy for $\alpha \in (7/8, 1]$, they require a utility (strictly) more than $3/2 = F_{\text{ag}_{i,1}}^{-1}(7/8)$. The fact that $v_1, v_2 \in [0,1]$ in addition to $v_1 + v_2 > 3/2$, implies that $v_1 > 1/2$ and $v_2 > 1/2$. Therefore, if a policy $\pi$ is $\alpha$-approved by agent $\text{ag}_{i,1}$, we have $\pi(\lambda(c_{i,1})) > 1/2$ and $\pi(\lambda(c_{i,2})) > 1/2$. Further, observe that for at most one of the three agents $\{\text{ag}_{i,k}\}_{k \in [3]}$ the condition of $J_{\text{ag}_{i,k}}(\pi) > 3/2$ may hold as every pair of the agent disagree on one literal.

We round such a policy $\pi$ to an assignment $\phi$ by letting $\phi(x_j) \leftarrow$ True if $\pi(a_{\text{True}}|s_j) > \frac{1}{2}$ and $\phi(x_j) \leftarrow$ False otherwise. If an agent in $\{\text{ag}_{i,k}\}_{k \in [3]}$ $\alpha$-approves $\pi$, then $C_i$ is satisfied by the assignment $\phi$. Therefore, we have that $\text{OPT}_\alpha \leqslant \text{OPT}_{\text{MAX}-2\text{SAT}}$, where $\text{OPT}_\alpha$ is the maximum feasible number of $\alpha$-approvals among all policies and $\text{OPT}_{\text{MAX}-2\text{SAT}}$ is the maximum number of clauses that can be satisfied among all assignments.

Next, we show $\text{OPT}_{\text{MAX}-2\text{SAT}} \leqslant \text{OPT}_\alpha$ by deriving a policy $\pi$ that gets $\text{OPT}_{\text{MAX}-2\text{SAT}}$ $\alpha$-approvals based on the optimal assignment for $\text{OPT}_{\text{MAX}-2\text{SAT}}$ by simply letting $\pi(a_{\text{True}}|s_j) = 1$ iff $\phi(x_i) = $ True. If $\phi$ satisfies a clause $C_i$, then for the agent $\text{ag} \in \{\text{ag}_{i,k}\}_{k \in [3]}$ that matches the literal assignments of $\phi$ for $C_i$, we have $J_{\text{ag}}(\pi) = 2$, which is an optimal, i.e., 1-approval, policy for ag. For the other two agents for $C_i$, $J_{\text{ag}}(\pi) = 1$ which gets a return less than their required utility of $3/2$ for an $\alpha$-approval. Therefore, we have that $\text{OPT}_{\text{MAX}-2\text{SAT}} = \text{OPT}_\alpha$ and the hardness of computation follows from our reduction. □

### B.3 Proof of Theorem 6

*Proof of Theorem 6.* The Borda count of a policy is defined as $\text{Borda}(\pi) = \sum_{i \in [n]} F_i(J_i(\pi))$. Since $F_i(v) \in [0,1]$, $\max_\pi \sum_{i \in [n]} F_i(J_i(\pi)) \leqslant n$. From Definition 6, for a $q$-quantile fair policy $\pi_q$, we have $F_i(J_i(\pi_q)) \geqslant q$, for all $i \in [n]$. Therefore, $\sum_{i \in [n]} F_i(J_i(\pi_q)) \geqslant qn$ and we have

$$\frac{\text{Borda}(\pi_q)}{\max_\pi \text{Borda}(\pi)} \geqslant \frac{qn}{n} \geqslant q. \qquad \square$$

## C   A Note on Algorithm 4

Here, we expand on Algorithm 4. As mentioned before, in the pre-processing step, for every agent $i$, we measure $F_i(k\epsilon)$ for $k \in [1/\epsilon]$ via $1/\epsilon$ calls to vol-comp. Let $J(\pi) = (J_1(\pi), \ldots, J_n(\pi))$ be the (expected) return vector. Importantly, our MILP approximates the Borda score of a return vector $r$ by its $\epsilon$-rounded-down return vector $r_\epsilon = (\lfloor r_i/\epsilon \rfloor \cdot \epsilon \mid i \in [n])$. Therefore, the MILP finds a policy that has a Borda count which is at least as high as the maximum Borda count among $\epsilon$-rounded return vectors. This is *not* necessarily equivalent to a $(1 - \epsilon)$ approximation of the max Borda count.

## D   Experimental Details

Experiments are all done on an AMD EPYC 7502 32-Core Processor with 258GiB system memory. We use Gurobi [18] to solve LPs and MILPs.

**Running time.** All our voting rules has a running time of less than ten minutes on the constructed MOMDPs of $3^5 = 243$ states, 6 actions, and 10 agents. The most resource extensive task of our experiments was sampling $5 \cdot 10^5$ random policies and computing the expected return for every agent which is a standard task. For each instance, we did this in parallel with 20 processes in a total running time of less than 2 hours per instance.

**Implementation details of Borda count rule.** After fitting a generalized logistic function for $F_i$ based on the expected return of sampled policies, we find the value $\text{mode}(f_i)$, and check the existence of a policy by solving a linear program that achieves a expected return of $\text{mode}(f_i)$ for all agents. Next, to utilize Gurobi LP solvers, we approximate the concave function $F_i$ by a set of linear constraints that form a piecewise linear concave approximation of $F_i$. Therefore, our final program for an approximate Borda count rule is simply an LP.

