# OpenReview forum: "Policy Aggregation"
_NeurIPS.cc/2024/Conference — NeurIPS 2024 poster_

### Official Review · Reviewer_61yS · 2024-07-11

**Soundness:** 3
**Presentation:** 3
**Contribution:** 2
**Rating:** 6
**Confidence:** 3

**Summary:**

This paper joins a long list of recent work that studies how to aggregate the preferences of several agents (e.g., humans) in a reinforcement learning framework inspired by social choice theory. The problem is modeled as a multi-objective MDP with $n$ different reward functions. The authors propose to use the state-action occupancy measure instead of each agent's most preferred policy or reward function directly. Popular concepts from social choice theory, such as the Borda count rule and approval voting, are then studied from this perspective.

**Strengths:**

- The paper is well-written and easy to read.
- It appears that considering the state-action occupancy measure has some advantages over working directly with each agent's optimal policy or reward function when attempting to introduce social choice rules, which---even though a standard approach in RL---is interesting.

**Weaknesses:**

- In my opinion, the contributions of this work are limited. E.g., only the full information case is studied
- The primary justification of this work (which is repeatedly mentioned in the paper) is that prior work on policy aggregation and fair RL is not invariant to affine transformations of the reward function. Essentially, agents can have differently scaled reward functions, which makes, e.g., maximizing for social welfare a bad objective. However, I don't understand why we cannot simply normalize the reward function of each agent, so that the reward functions are directly "comparable". I find the concern about affine transformations quite weak.

**Questions:**

- I'm a bit surprised about the title of the paper since you're not aggregating policies, but reward functions. In fact, the policies play a minor role in the paper, since you look at the preference relation induced by the reward function (which you then express in term of occupancy measures). Could you explain why what you're doing is policy aggregation and not just preference aggregation?

Typo: "Policy Aggergation" in title of Section 5

**Limitations:**

Limitations are adequately addressed in my opinion.

---

> ### Author Rebuttal · Authors · 2024-08-06
>
> > The primary justification of this work (which is repeatedly mentioned in the paper) is that prior work on policy aggregation and fair RL is not invariant to affine transformations of the reward function. Essentially, agents can have differently scaled reward functions, which makes, e.g., maximizing for social welfare a bad objective. However, I don't understand why we cannot simply normalize the reward function of each agent, so that the reward functions are directly "comparable". I find the concern about affine transformations quite weak.
>
> We disagree with this comment and believe we can provide an effective rebuttal. As we mention in lines 34-37, there is a rich literature in economics about the shortcomings of interpersonal comparison of utility. Note that there is nothing about our setting that makes normalization of utilities especially compelling, so your proposed solution is equally relevant in every setting that involves interpersonal comparison of utility. If normalization "worked," therefore, interpersonal comparison of utility would be a nonissue.
>
> There are several arguments against normalizing utilities, but instead of a philosophical discussion, perhaps an example would be most useful. Suppose there are three alternatives, agent 1 has utilities (0, 1/2, 1) and agent 2 has utilities (2, 3, 4). Should the normalized utilities of agent 2 be (1/2, 3/4, 1), or should we first subtract 2 and then divide by the maximum, yielding (0, 1/2, 1)? Or should we only subtract 1? Or perhaps we should normalize so that the sum of utilities is equal, giving (0, 1/3, 2/3) for agent 1 and (2/9, 1/3, 4/9) for agent 2? Economists argue that there is no principled way to answer these questions.
>
> > I'm a bit surprised about the title of the paper since you're not aggregating policies, but reward functions. In fact, the policies play a minor role in the paper, since you look at the preference relation induced by the reward function (which you then express in term of occupancy measures). Could you explain why what you're doing is policy aggregation and not just preference aggregation?
>
> The way we imagine the eventual pipeline is that we would observe trajectories from the optimal policy of each agent, learn reward functions via inverse reinforcement learning, and apply our techniques to select a policy. In this sense, we're aggregating the individual optimal policies into a collective policy. We also like the use of the word "policy" as it suggests that we're dealing with MDPs, in contrast to "preference aggregation," which is much more general. That said, we'd certainly be open to changing the title if this is seen as a pivotal issue.

---

> > ### Comment · Reviewer_61yS · 2024-08-12
> >
> > Thank you for your response and sorry for my late reply. My concerns are addressed and I can align with the other reviewers on a score of 6. Regarding the title, in my opinion, something along the lines of “socially fair reward aggregation” would better capture the point of the paper. I think, it’s fine either way, and it seems that I was the only one who expected slightly different content given the title (I was expecting direct policy aggregation).

---

### Official Review · Reviewer_7MP7 · 2024-07-12

**Soundness:** 3
**Presentation:** 3
**Contribution:** 3
**Rating:** 6
**Confidence:** 3

**Summary:**

The paper solves the problem which arises in preference aggregation of individual policies to a collective policy – (1) summation based aggregation are sensitive to affine transformations and (2) voting rule based aggregation faces problem of policies being exponential in S. Towards solving this, the paper proposes voting over continuous space of alternatives (which eliminates affine sensitivity) and a volumetric definition of preference ordering. The paper next proposes efficient algorithms to (1) find approximate volumetric veto core and (2) approximate q-Quantile Fairness. They also show complexity of existing voting rules, notably plurality voting and borda count. They show that problem is computationally hard for plurality and open for broad count. I am inclined towards accepting the paper.

**Strengths:**

The paper solves a well-motivated problem of policy aggregation. Their proposal of achieving different notions of approximate fairness through efficient algorithms both novel and appears to be sound. Their theoretical analysis of the complexity of using plurality and borda count based voting is also significant and allows scope for future work in the direction. Their algorithms have been validated through experiments.

**Weaknesses:**

In the experimental section, using a common metric to quantify the “level of fairness” guaranteed by different algorithms would be beneficial for a more learned comparison.

In Def. 4 should the expression be vol(O’)/vol(O) >= 1 – veto(S) + epsilon instead of vol(O’) >= 1 – veto(S) + epsilon as currently stated?

**Questions:**

see weakness.

Why can't yours be a special case of Noothigattu et al. [27]?

**Limitations:**

Authors adequately address the problem statement and discussed limitations/candidate improvements of their work which is left to future work.

---

> ### Author Rebuttal · Authors · 2024-08-06
>
> > In Def. 4 should the expression be vol(O’)/vol(O) >= 1 – veto(S) + epsilon instead of vol(O’) >= 1 – veto(S) + epsilon as currently stated?
>
> Yes, you are right. Thank you for catching this typo.
>
> > Why can't yours be a special case of Noothigattu et al. [27]?
>
> Noothigattu et al. assume that there's a ground-truth reward function and the agents' reward functions are random perturbations thereof. Consequently, they essentially propose to treat all the data as coming from a single agent and directly apply an inverse reinforcement learning algorithm to the pooled data. Our setup, by contrast, calls for pluralistic rules that take diverse preferences into account.
>
> Perhaps a good analogy is the shift that has been happening in the last year from reinforcement learning from human feedback (RLHF) methods that rely on the Bradley-Terry model and essentially assume there is one type of person to "pluralistic alignment" methods where social choice plays a major role and diverse preferences are respected. See these recent position papers for more details:
>
> Taylor Sorensen, Jared Moore, Jillian Fisher, Mitchell Gordon, Niloofar Mireshghallah, Christopher Michael Rytting, Andre Ye, Liwei Jiang, Ximing Lu, Nouha Dziri, Tim Althoff, Yejin Choi. A Roadmap to Pluralistic Alignment. arXiv:2402.05070, 2024.
>
> Vincent Conitzer, Rachel Freedman, Jobst Heitzig, Wesley H. Holliday, Bob M. Jacobs, Nathan Lambert, Milan Mossé, Eric Pacuit, Stuart Russell, Hailey Schoelkopf, Emanuel Tewolde, William S. Zwicker. Social Choice Should Guide AI Alignment in Dealing with Diverse Human Feedback. arXiv:2402.05070, 2024.
>
> > In the experimental section, using a common metric to quantify the “level of fairness” guaranteed by different algorithms would be beneficial for a more learned comparison.
>
> The current plots show the utility of different quantiles (10%, 20%, ..., 100% for the experiment with 10 agents and 20%, 40%, ..., 100% for the one with 5 agents), which provides a holistic view of the utility distribution. The bars for the worst-off agent correspond to the egalitarian welfare. In the revised manuscript, we will include additional plots with information on different fairness metrics such as the Gini coefficient and Nash welfare. If the reviewing team also has other suggestions, we would gladly consider them.

---

### Official Review · Reviewer_vrsi · 2024-07-16

**Soundness:** 4
**Presentation:** 4
**Contribution:** 3
**Rating:** 7
**Confidence:** 3

**Summary:**

This paper studies aggregating multiple policies–which can be seen as a formalization of the task of aligning an AI system to the values of multiple individuals. When the number of states is small (such as, when multiple individuals have to select one out of a few candidates), this problem has been widely studied in voting and social choice theory, and there are many efficient aggregation rules (such as the Borda count). This paper, however, considers the other extreme: where the state-action space is huge, and it is not obvious how to design efficient methods to aggregate policies.

The main insight of this paper is that preferences over policies have a volumetric interpretation in the state-action policy space that, in some cases, leads to efficient aggregation algorithms.

Concretely, the authors examine two types of aggregation rules: (1) two aggregation rules that are known to have desirable fairness properties (namely, proportional veto code and, the recently introduced, quantile fairness) and (2) voting or score-based rules such as Borda count and $\alpha$-approval voting rule.

Building on their insight the authors prove several results, including
1. an algorithm which finds the policy wrt an $\epsilon$-approximation of the proportional veto core using $O(log(1/\epsilon))$ queries to a volume computation oracle,
2. the existence of $q$-quantile fair policies for all $q\geq 1/e$ (which is tight and stronger than the best possible bound in the discrete case),
3. NP-hardness and inapproximability results for $\alpha$-approval score.

**Strengths:**

The paper is well-written and easy to read. I believe that the problem proposed in the paper is well-motivated from alignin AI systems, and is of significant interest to research on voting rules and social choice theory. The theoretical results are solid and the proofs and/or approach are well outlined. Finally, I did not check the proofs in detail, but they appear sound. One caveat is that I am not familiar with the closely related prior work (e.g., [6]) and, so, cannot comment on the novelty of the proofs and results from prior work.

**Weaknesses:**

I am not sure if the empirical results section is adding any value to this paper: it evaluates different aggregation rules, but I think this is not the focus of this work–I think the focus is to design efficient algorithms and/or prove existential results. If other reviews and the area chairs agree, my suggestion is to drop the empirical results section and use the additional space to add more exposition on the proofs. To be clear, this is no a significant concern for me.

**Questions:**

I do not have any specific questions for the authors.

**Limitations:**

I do not see significant limitations.

---

> ### Author Rebuttal · Authors · 2024-08-06
>
> > I am not sure if the empirical results section is adding any value to this paper: it evaluates different aggregation rules, but I think this is not the focus of this work–I think the focus is to design efficient algorithms and/or prove existential results. If other reviews and the area chairs agree, my suggestion is to drop the empirical results section and use the additional space to add more exposition on the proofs. To be clear, this is no a significant concern for me.
>
> We agree that the main contribution and focus of our work is theoretical, that is, efficient algorithm design and existential results as mentioned. We are open to moving the empirical section to the appendix and elaborating on the theory, including the proofs, in the main body.
>
> We do see the empirical results as contributing in two ways: (1) The aggregation rules are practical. They can be easily implemented and run fast (on not-so-small instances). (2) In addition to theoretical arguments, empirical evaluation illustrates the differences in policies generated by the proposed rules, e.g.,  some of the proposed rules result in policies that are empirically "fairer" to different stakeholders in complex settings.

---

> > ### Comment · Reviewer_vrsi · 2024-08-12
> > **Response to authors**
> >
> > I thank the authors for their response. I will keep my previous score and think this paper should be accepted.

---

### Decision · Program_Chairs · 2024-09-25

**Decision:**

Accept (poster)

**Comment:**

Reviewers are all positive about the paper. They consider the paper to be well-motivated from aligning AI systems and they find the paper to be well-written. It will be good to address reviewers' comments on the empirical section, and it might be good to add the discussion about normalizing utilities in the rebuttal to the paper if there's space.